# Perceptions of green space usage, abundance, and quality of green space were associated with better mental health during the COVID-19 pandemic among residents of Denver

**Colleen E. Reid**[1,2]*, **Emma S. Rieves**[1,2], **Kate Carlson**[1]

**1** Geography Department, University of Colorado Boulder, Boulder, Colorado, United States of America,
**2** Institute of Behavioral Science, University of Colorado Boulder, Boulder, Colorado, United States of America

* Colleen.Reid@colorado.edu

**Data Availability Statement:** Our data was approved and accepted by CU Scholar at the

## Abstract

### Background

The COVID-19 pandemic has impacted both physical and mental health. This study aimed to understand whether exposure to green space buffered against stress and distress during the COVID-19 pandemic while taking into account significant stressors of the pandemic.

### Methods

We leveraged a cross-sectional survey on green space exposure and mental health among residents of Denver, CO that ran from November 2019 through January 2021. We measured objective green space as the average NDVI (normalized difference vegetation index) from aerial imagery within 300m and 500m of the participant's residence. Perceived green space was measured through Likert scores on five questions about vegetation near the home that captured perceived abundance, visibility, access, usage, and quality of green space. We used generalized linear models to assess the relationship between each green space exposure variable and perceived stress (PSS-4), depression (CES-D-10), or anxiety (MMPI-2) adjusted for sociodemographic and COVID-19 impact variables.

### Results

We found significantly higher depression scores for all covid periods compared to the "before covid" period, and significantly higher anxiety scores during the "fall wave" compared to earlier periods. Adjusted for sociodemographic and pandemic stressors, we found that spending a lot of time in green space (usage) was significantly associated with lower anxiety and depression. We also observed significantly lower depression scores associated with NDVI in both buffers (objective abundance) and significantly lower anxiety scores with perceived abundance of green space. There was some evidence of lower anxiety scores for

University of Colorado Boulder and can be found here: https://doi.org/10.25810/3wjs-ay43.

**Funding:** Research reported in this publication was supported by the Harvard JPB Environmental Health Fellowship (CER) and the Developmental Core of the University of Colorado Population Center (CUPC) (CER). The CUPC is supported by funds from the Eunice Kennedy Shriver National Institute of Child Health & Human Development of the National Institutes of Health (NIH) under Award Number P2CHD066613. The content is solely the responsibility of the authors and does not necessarily represent the official views of the NIH. NIH and CUPC did not play a role in the study design, data collection and analysis, decision to publish, or preparation of the manuscript. Researchers at Harvard provided feedback on the study design, but the JPB funders did not. Neither JPB nor Harvard played a role in the data collection and analysis, decision to publish, or preparation of the manuscript.

**Competing interests:** The authors have declared that no competing interests exist.

people reporting having high quality green spaces near the home (quality). We did not observe significant associations for any green space metric and perceived stress after adjustment for confounding variables.

## Conclusion

Our work provides further evidence of mental health benefits associated with green space exposure during the COVID-19 pandemic even after adjustment for sociodemographic variables and significant pandemic-related stressors.

## Introduction

Since it was first recognized over a year ago in Wuhan, China, a disease––referred to as COVID-19––caused by a novel coronavirus, SARS-CoV-2, has led to almost 185 million cases and over 4 million deaths worldwide (https://coronavirus.jhu.edu/map.html on July 9, 2021). In this time, it has been widely acknowledged that the COVID-19 pandemic has not just caused physical health but also mental health concerns [1–7]. This is true not only for those infected with the SARS-CoV-2 virus, but also for health care workers and the general public [2]. Indeed, multiple studies and systematic reviews have already documented increases in mental health impacts––such as depression, anxiety, and stress––associated with the pandemic among the general public [5]. The causes of the mental health impacts are likely numerous, including stress and distress from the loss of loved ones, the existential toll of COVID-19-related morbidity and mortality among populations [6, 8], anxiety and fear about contracting a virus with potential long-term health implications, not being able to work from home and thus risking potential exposure [1], anxiety and depression related to financial concerns and job insecurity [5, 9], and the mental health toll from increased isolation due to lockdowns, stay-at-home orders, or intermittent quarantines [3, 4, 10]. Increased social media exposure and media consumption might also exacerbate mental health concerns, especially given the high volume of contradictory information and misinformation about COVID-19 [1, 5, 11].

In March of 2020, many states imposed a variety of social distancing measures to try to curb the spread of SARS-CoV-2, with varying levels of strictness by governments and varying levels of compliance by populations. Most schools and non-essential workplaces were closed, resulting in many adults and children working or attending school from home and an increased use of computers and screen time. To increase compliance with social distancing measures, many local and state governments closed indoor places of recreation, including gyms. Interestingly, the closure of indoor places of recreation may have led to more people taking walks, often in parks and nature trails, as their only means of exercise. In other locations, use of green space declines due to some cities initially closing many parks and natural areas [12]. Given the growing evidence of numerous health benefits–from increased mental health, decreased birth outcomes, less cardiovascular morbidity, increased longevity, and more—from exposure to natural vegetation in urban environments, often called "green space" or greenness [13], it is important to understand whether green space exposure during the pandemic impacted people's mental health.

Indeed, more and more literature documents how people's interactions with green space changed during the COVID-19 pandemic. Some studies report decreased levels of green space exposure during the pandemic [14–16], whereas others report that the duration and frequency of green space visits increased during the pandemic [17–20]. Evidence indicates that

interactions with green spaces might vary based on the stringency of lockdown policies, socioeconomic status, and work flexibility [16, 17] how far away the green space is [21, 22], or potentially due to differences in lockdown policies in different places [19, 23].

Not only have studies documented changes in usage of green space during the pandemic, but also whether exposure to vegetation has affected mental health during the COVID-19 pandemic. Studies have found that access to views of green space, visitation to urban parks and public spaces, access to private green spaces, neighborhood greenery [14, 23–25] and increased duration and frequency of green space exposure [20] have been associated with decreased stress and distress during the pandemic. Exposure to indoor green space, such as having houseplants, has also been associated with better mental health during the pandemic [20, 23, 25] and ongoing research is exploring how virtual green space during the pandemic is associated with stress, depression, and anxiety [26] feelings of connectedness and decreased loneliness [27].

Studies examining the relationships between green space and health assess exposure by using either researcher-generated "objective" estimation of green space exposure or self-reported, or perceived, exposure to green space from the study participant. Objective green space exposures normally consists of intersecting areal units around where someone lives (i.e., census tract, ZIP code, or radial buffer around address) with measures of vegetation exposure calculated from satellite data or land use or land cover data (e.g., the Normalized Difference Vegetation Index (NDVI) or tree canopy coverages) [13]. Perceived or self-reported exposure to green space normally entails asking questions in a survey about various aspects of green space exposure related to its quality [28–30], access [31], safety [32, 33], abundance [34], proximity [35], or the participants' frequency and duration of use [36]. Previous research has shown that perceived and objective exposures do not often align [37–41] and some hypothesize that perceptions of green space exposure may matter more than objective measures for certain pathways by which green space is hypothesized to affect health [29].

Our study leverages a survey on green space exposure and mental health that began in the winter of 2020 among residents of Denver, CO and continued into 2021. In May of 2020, we added questions about the impacts of the COVID-19 pandemic and associated social distancing and stay at home orders. Utilizing a survey that was already collecting responses allows our investigation to address some important questions about green space and mental health differently from other studies: (1) we can compare––from the same population––metrics of stress, depression, and anxiety from multiple time periods during the pandemic, (2) we have information on a comprehensive list of impacts of the pandemic (related to finances, health, and access to resources) that may relate to stress, depression, and anxiety that we can control for, and (3) our questions about impacts of the pandemic come at the end of the survey and do not directly address mental health and green space exposures together such that respondent is not asked a question that directly relates to our hypotheses. Additionally, our survey contains questions that allow us to quantify objective green space exposure and compare it to multiple dimensions of perceived green space exposure including abundance, visibility, access, usage, and quality. We hypothesize that 1) perceived stress, depression and anxiety metrics increased during the pandemic in Denver compared to before, and that 2) green space exposure is associated with better mental health metrics during the pandemic even when adjusted for sociodemographic confounders and stressful impacts of the COVID-19 pandemic.

## Methods

### Study population

We recruited participants from neighborhoods within Denver, CO, which is a consolidated city and county, for a study on the relationship between green space and health. Denver is an

interesting city from which to study green space exposure and health because of previous studies documenting unequal access to parks by race/ethnicity in Denver [42] and lower levels of vegetation, as measured by satellites, in lower income areas of Denver [43].

For our survey population, we targeted neighborhoods with differing levels of greenness and median income levels with the intent of decoupling in our sample the known relationships between urban green space and socio-economic status [44, 45]. Participation in the survey was limited to respondents who were 18 years of age, lived within Denver at the time of taking the survey, and consented to be a part of the research. Participants were given a $20 electronic gift card as an incentive to participate in the survey. We recruited survey participants through postal and electronic mailings.

## Time period definition

The governor of Colorado issued a "Declaration of a Disaster Emergency related to COVID-19" on March 10, 2020. The World Health Organization declared a pandemic for COVID-19 infections on March 11, 2020. The mayor of Denver declared a state of emergency and the Denver Public Schools closed on March 12, 2020, because of concerns about the COVID-19 pandemic. Denver issued a "stay-at-home" order for the city on March 23, 2020. Given that life began to change significantly for residents of Denver on March 12––more than a week before the official "stay-at-home" order was enacted––we defined that as the last day of the "before covid" period. Denver officially ended its "stay-at-home" order and moved to Colorado's "safer at home" designation on May 9, 2020. Denver County experienced a precipitous rise in COVID-19 cases in the fall of 2020 even though public health orders did not significantly alter the "safer at home" order at that time. October 12, 2020, was the first day that daily cases in Denver County exceeded 200, and also marked the beginning of a swift upward trend in daily new cases. Headlines from October 13th's Denver Post described Colorado as facing a "third wave" marked by a state positivity rate exceeding 5% and hospitalizations reaching previous highs. For this study, we will use survey results from before March 13 as from the "before covid" period, results from March 13 to May 8 as from the "stay at home" period, results from May 9 to October 13, 2020, as the "reopening" time period, and results from October 13 to January 2, 2021 as the "fall wave".

## Geocoding of participant locations

We cleaned, standardized, and then geocoded location information (home address or nearest intersection to the home) provided by survey participants with ESRI's ArcGIS World Geocoding Service Address Locator [46] using the Denver street centerline and parcel data.

## Measures of stress and mental health

Our survey used the four-item perceived stress scale (PSS-4) adapted from the larger 14-item perceived stress scale created by Cohen and colleagues [47]. These four questions ask about how often in the past month the person felt (1) unable to control things in their life, (2) confident to handle their personal problems, (3) that things were going their way, and (4) that things were piling up. Respondents could choose among five categories for each of the four responses on a Likert scale of "never", "almost never", "sometimes", "fairly often", and "very often". We scored answers from 0 to 4, with reverse coding for questions 2 and 3, such that the overall summed scale ranged from 0 to 16 with higher values indicating more stress.

We used the 10 question version of The Center for Epidemiological Studies Depression Scale (CES-D-10) [48] that previous studies have found was equivalent to the original 20 questions, but faster to answer [49]. The statements in this scale relate to whether, during the past

week, the person is bothered by things that normally do not bother them, they have trouble keeping their mind on what they were doing, they felt depressed, whether things feel like an effort, they feel hopeful about the future, they felt fearful, their sleep was restless, they felt happy, they felt lonely, or they could not "get going". Participants chose between never (0), rarely (0), sometimes (1), fairly often (2), or very often (3) for each question. The answers were scaled from 0 to 3 with the positively-worded statements reverse-coded so that higher scores represent an increased likelihood of depression. Scores can range from 0 to 30, but there is no agreed upon threshold for labeling a respondent as depressed across populations [50]; the tool should not be used for diagnostic purposes but rather as a way to assess symptom severity for depression [51].

We used the 23-item Minnesota Multiphasic Personality Inventory-2 Anxiety Scale (MMPI-2 Anxiety) to assess symptoms of anxiety in our survey [52]. These 23 questions ask about participant nervousness, focus, sleep, stress, and anxiety. Responses are based on how frequently a statement applies to the respondent, with the scale ranging from (1) rarely or none of the time, (2) some or little of the time, (3) a moderate amount of time, to (4) most or all of the time [52]. All responses were scored from 1 to 4 with some statements reverse coded such that higher scores indicate higher symptoms of anxiety.

On May 12, 2020, we added questions to our survey related to how the COVID-19 pandemic affected respondents. These questions were taken from the Social Psychological Survey of COVID-19: Coronavirus Perceived Threat, Government Response, Impacts, and Experiences Questionnaires (https://www.phenxtoolkit.org/covid19) [53]. These questions asked (1) whether the pandemic has impacted the person financially, (2) if the respondent had lost any job-related income due to COVID-19, (3) if they had a hard time finding needed resources (such as food and toilet paper), (4) if they had to keep working in close contact with others during this time, (5) if they were ever diagnosed with COVID-19, (6) if they had ever experienced symptoms of COVID-19, (7) if they had been in close proximity to someone who was diagnosed or had symptoms of COVID-19, and (8) whether they spent a large amount of their time trying to find information about COVID-19 from TV and the internet. We also added in our own question (9) about whether the respondent was using green space more or less now than during the same time period in the previous year. Respondents responded to each of these questions with a value between 1 and 7 where 1 = "not true of me at all" and 7 = "very true of me". We recoded the question about having been diagnosed with COVID-19 as binary; all but one respondent answered either a 7 or a 1 for that question.

We also collected respondents' demographic information, including: their gender, age, race/ethnicity, income group, marital/partnered status, employment status, educational attainment, and health insurance status. We re-coded respondents who chose more than one race category as multiracial. For health insurance status, we re-coded the multiple health insurance types into whether someone had any form of health insurance (e.g., Medicare, Medicaid, insurance from military service, or private insurance) or not. Our question on employment allowed for multiple responses and also for choices such as "out of work and looking for work" and "out of work and not looking for work". As we were most interested in whether people were looking for work or not, we classified anyone who indicated that they were looking for work as "looking for work" and those that indicated that they were employed part-time as "looking for work part-time". Anyone who did not indicate that they were "looking for work" was classified as "not looking" for work. This category included people who indicated that they were not looking for work but were employed full time, retired, a student, a homemaker, out of work, or unable to work.

The survey was made available to respondents in Spanish or English on Qualtrics. The survey questions are available in the S1 File.

## Measures of perceived green space exposure

Our survey asked questions about respondents' perceptions of their exposure to green space. These questions asked participants to what extent they agree (with the options of strongly disagree, disagree, agree, and strongly agree) with five statements about presence of natural vegetation/green space in their neighborhood. The statements were: "There is a lot of vegetation/greenery in my neighborhood", "I can see vegetation/greenery from my home", "The nearest vegetated park/green space is easy for me to access", "I spend a lot of time in spaces with natural vegetation", and "The green spaces near my home are very high in quality". These statements were adapted from perceived green space questions in Dzhambov et al. [29]. The prompt for these statements includes a definition of "green space" as "any area with natural vegetation. This can include parks, yards, grassy areas, street trees, green roofs, cemeteries, etc." We investigated these metrics as separate measures of perceived green space exposure instead of as a composite in this study because we were interested in comparing the different aspects of green space exposure that they convey: abundance, visibility, access, usage, and quality.

## Objective measures of green space exposure

We used recent (2019) aerial imagery from the National Agriculture Imagery Program (NAIP) at 1m spatial resolution to calculate the Normalized Difference Vegetation Index (NDVI), a metric commonly used in studies of the health impacts of objective green space exposure [13]. The NAIP data provide high resolution (1m) aerial imagery taken with red, green, blue, and near infrared bands during summer leaf-out. The NDVI is calculated as the difference between the near infrared and red bands divided by the sum of those two bands because, compared to non-vegetated surfaces, green vegetation absorbs more energy in the red wavelength and reflects more infrared radiation compared to non-vegetated surfaces [54]. NDVI values range from -1 to 1 with higher values corresponding to healthy vegetation and values below 0 representing areas with no vegetation. Before averaging NDVI values, we removed all negative values as these do not represent vegetation and would mathematically offset the values above one that do. We determined each participant's green space exposure as the average of positive NDVI values within 300- and 500-meter radial buffers around each survey respondent's geocoded address or nearest cross-street. There is no agreement on what radial buffer size is most appropriate in green space and health studies. Different studies have found that both larger and smaller buffers are better, but despite this disagreement, the clearest evidence is that the buffer size matters [55, 56]. We chose these buffer sizes because they are commonly used in green space and mental health research [57].

## Statistical analysis

We used pairwise Wilcox tests [58] to assess if PSS-4, CES-D-10 depression, and MMPI-2 anxiety scores significantly changed across the population during four time periods: "before covid", "stay at home", "reopening", and the "fall wave".

To understand to what extent the COVID-19 impacts explained the PSS-4, CES-D-10, or MMPI-2 anxiety scores among our survey population, we performed univariate generalized linear models (GLMs) for each COVID-19 experience variable (diagnosed with COVID, symptoms of COVID, impacted financially, lost income, hard time getting resources, had to work in close contact with potentially infected individuals, being in close proximity to someone experiencing symptoms or diagnosed with COVID, and spending a huge amount of time online or on TV trying to find out information about COVID-19) with each of the three stress measures (PSS-4, CES-D-10, and MMPI-2 anxiety). With the exception of the COVID-19

diagnosis variable–which we coded as binary–we retained the continuous coding for the COVID-19 impact variables, such that an increasing value indicates a higher experienced impact.

To evaluate the role of green space in moderating the impacts of the COVID-19 pandemic on stress and mental health, we first evaluated whether perceived green space exposures (each of the five questions on perceived green space) and objective green space exposures (NAIP NDVI at 300m and 500m radial buffers) significantly predicted Cohen's PSS-4, CES-D-10 depression, and MMPI-2 anxiety scores in univariate linear regression models. As green space is often confounded by sociodemographic factors, we then evaluated whether the associations in the univariate analyses remained after adjustment for the set of sociodemographic variables that significantly affected each outcome variable (Cohen's PSS-4, CES-D-10 depression, and MMPI-2 anxiety) based on statistical significance of each variable (i.e., sex, race, ethnicity, income group, having a Bachelor's degree or higher, currently having a partner in one's life, having health insurance, looking for work) with each outcome variable in univariate regressions (S1 Table). This means that each outcome variable had a slightly different list of sociodemographic adjustment variables in these multiple regression analyses. In our third set of models, we additionally adjusted for COVID-19 impact variables that were significant in univariate regression with that outcome variable. We performed VIF statistics and found no collinearity in any of these regression models.

All objective green space data processing, address geocoding, radial buffer creation, and objective green space exposure assignment was done using ESRI's ArcGIS version 10.8.1 [59]. All statistical analyses were done using R version 4.0.5 [60]. This study was approved by the Institutional Review Board of the University of Colorado Boulder with protocol number 19–0429.

## Results

### Study population

We received 1188 survey responses between November 16, 2019, and January 2, 2021; 912 of these survey responses were considered complete enough for analysis (i.e., greater than 50% of the survey completed). This represents a response rate of 6.2% and a completion rate of 4.8%. Of the 912 completed responses, one response was dropped from this analysis because the respondent did not respond to any demographic questions. We dropped eight more observations because they could not be assigned objective green space exposures due to insufficient geographic information provided in their survey responses. Of the remaining 903, 94 (10%) of these responses occurred during the "before covid" period, 309 (34%) responded during the "stay-at-home" period, 201 (22%) responded during the "reopening" period, and 299 (33%) responded during the "fall wave" period. Fig 1 depicts the number of observations used in each analysis.

Respondents to our survey were more likely to be female, older, White, and not Hispanic/Latino than the population of Denver or the neighborhoods in Denver that we targeted. They were also more likely to have a Bachelor's degree, and a higher income than the average in Denver or the neighborhoods we targeted, but they were about equally likely to have health insurance as the population of Denver (S2 Table).

### Descriptive statistics

Descriptive statistics for survey respondents used in our analyses and by period are shown in Table 1. 88% of survey respondents had health insurance, 73% had a bachelor's degree or higher (73%), 86% were not looking for work and 58% were female. The majority of

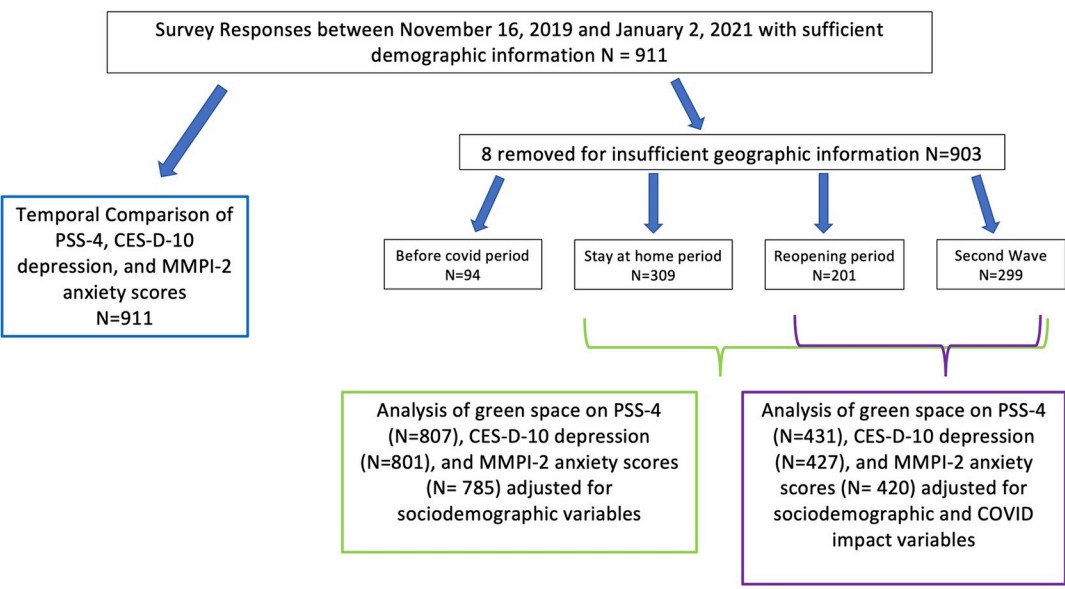

**Fig 1. Flow chart of survey responses used in each analysis.**

respondents were in higher income groups and people aged 30–49 responded at higher rates than other age groups. We had more responses from individuals who identified as White (89%) and as not Hispanic/Latino (86%) than other racial and ethnic groups.

Although only 2.6% of respondents reported having been diagnosed with COVID-19, 8.1% of our respondents reported experiencing COVID-19-like symptoms at some point, 18% reporting having been in close contact with someone who was diagnosed with COVID-19, 17% reported having been in close contact with someone who had COVID-19-like symptoms, and 22% reported having to continue to work even their employment put that at greater risk of being exposed to someone infected with SARS-CoV-2 (S3 Table). More respondents were affected financially by the pandemic in our sample than were potentially exposed to the virus with 31% of respondents reporting that they lost job-related income due to the pandemic, and 35% saying that they were impacted financially by the pandemic. 13% of respondents reported higher levels of having trouble finding necessary supplies due to the pandemic, and 13% reported spending a lot of time online or watching TV trying to find out information about COVID-19. 33% of respondents reported spending more time in parks, trails and near nature than at the same time the year before.

We observed moderately positive Spearman correlations between perceived measures of green space exposure (r = 0.48 to 0.78). The range of correlations is likely due to the fact that the perceived green space questions were not all intended to describe the same aspects of green space exposure. The highest correlation was between the measure of nearby vegetation abundance ("There is a lot of vegetation/greenery in my neighborhood") and nearby green space visibility ("I can see vegetation/greenery from my home"). This relationship makes sense as an abundance of green space makes it likely that at least some of it is visible from one's home. The lowest correlation was between access ("The nearest vegetated park/green space is easy for me to access") and usage ("I spend a lot of time in spaces with natural vegetation").

Perceived and objective measures of green space exposure had Spearman correlations between 0.30 and 0.59, with the lowest correlation between NAIP NDVI in a 500 meter buffer and the question about access ("The nearest vegetated park/green space is easy for me to

**Table 1. Sociodemographic summary statistics for survey respondents.**

| Sociodemographic variables | Total N = 911[1] | COVID-19 time period | | | |
|---|---|---|---|---|---|
| | | Before COVID-19 N = 95 (10%)[1] | Stay at home period N = 313 (34%)[1] | Reopening period N = 203 (22%)[1] | Second wave N = 300 (33%)[1] |
| Sex | | | | | |
| Female | 526 (58%) | 57 (60%) | 171 (55%) | 122 (60%) | 176 (59%) |
| Male | 385 (42%) | 38 (40%) | 142 (45%) | 81 (40%) | 124 (41%) |
| Age | | | | | |
| 18 to 29 | 186 (20%) | 11 (12%) | 61 (19%) | 34 (17%) | 80 (27%) |
| 30 to 49 | 390 (43%) | 35 (37%) | 129 (41%) | 89 (44%) | 137 (46%) |
| 50 to 64 | 169 (19%) | 24 (25%) | 61 (19%) | 45 (22%) | 39 (13%) |
| 65+ | 166 (18%) | 25 (26%) | 62 (20%) | 35 (17%) | 44 (15%) |
| Income | | | | | |
| Less than $25,000 | 98 (11%) | 5 (5.3%) | 23 (7.5%) | 29 (15%) | 41 (14%) |
| $25,000 to $50,000 | 148 (17%) | 15 (16%) | 53 (17%) | 29 (15%) | 51 (17%) |
| $50,000 to $75,000 | 140 (16%) | 18 (19%) | 37 (12%) | 29 (15%) | 56 (19%) |
| $75,000 to $100,000 | 135 (15%) | 16 (17%) | 47 (15%) | 24 (12%) | 48 (16%) |
| $100,000 to $150,000 | 181 (20%) | 17 (18%) | 74 (24%) | 38 (19%) | 52 (18%) |
| Greater than $150,000 | 191 (21%) | 23 (24%) | 71 (23%) | 50 (25%) | 47 (16%) |
| Missing | 18 | 1 | 8 | 4 | 5 |
| Ethnicity | | | | | |
| Not Hispanic/Latino | 779 (86%) | 79 (83%) | 275 (88%) | 168 (83%) | 257 (86%) |
| Hispanic/Latino | 132 (14%) | 16 (17%) | 38 (12%) | 35 (17%) | 43 (14%) |
| Race | | | | | |
| White | 780 (89%) | 89 (95%) | 268 (88%) | 169 (88%) | 254 (89%) |
| Asian/Pacific Islander | 16 (1.8%) | 2 (2.1%) | 7 (2.3%) | 4 (2.1%) | 3 (1.0%) |
| Black/African American | 33 (3.8%) | 1 (1.1%) | 12 (4.0%) | 6 (3.1%) | 14 (4.9%) |
| Multiracial | 29 (3.3%) | 2 (2.1%) | 12 (4.0%) | 7 (3.6%) | 8 (2.8%) |
| Native American | 18 (2.1%) | 0 (0%) | 4 (1.3%) | 6 (3.1%) | 8 (2.8%) |
| Missing | 35 | 1 | 10 | 11 | 13 |
| Cohabitation status | | | | | |
| Partnered | 443 (49%) | 43 (45%) | 162 (52%) | 104 (51%) | 134 (45%) |
| Unpartnered | 468 (51%) | 52 (55%) | 151 (48%) | 99 (49%) | 166 (55%) |
| Educational attainment | | | | | |
| BA or higher | 666 (73%) | 77 (82%) | 234 (75%) | 145 (71%) | 210 (70%) |
| Less than BA | 244 (27%) | 17 (18%) | 79 (25%) | 58 (29%) | 90 (30%) |
| Missing | 1 | 1 | 0 | 0 | 0 |
| Looking for work? | | | | | |
| Not looking | 786 (86%) | 81 (85%) | 278 (89%) | 176 (87%) | 251 (84%) |
| Looking—part-time | 67 (7.4%) | 9 (9.5%) | 21 (6.7%) | 12 (5.9%) | 25 (8.3%) |
| Looking | 58 (6.4%) | 5 (5.3%) | 14 (4.5%) | 15 (7.4%) | 24 (8.0%) |
| Insurance status | | | | | |
| Insured | 806 (88%) | 86 (91%) | 282 (90%) | 187 (92%) | 251 (84%) |
| Not insured | 105 (12%) | 9 (9.5%) | 31 (9.9%) | 16 (7.9%) | 49 (16%) |

[1]Before COVID-19 (before 3/12/21); Stay at home (3/13/20-5/8/20); Reopening (5/9/20-10/12/20); Second wave (10/13/20-1/2/21)

access") and the highest between NAIP NDVI (within either radial buffer) and the question about green space abundance ("There is a lot of vegetation/greenery in my neighborhood"), as would be expected given that NDVI is a measure of amount of vegetation nearby.

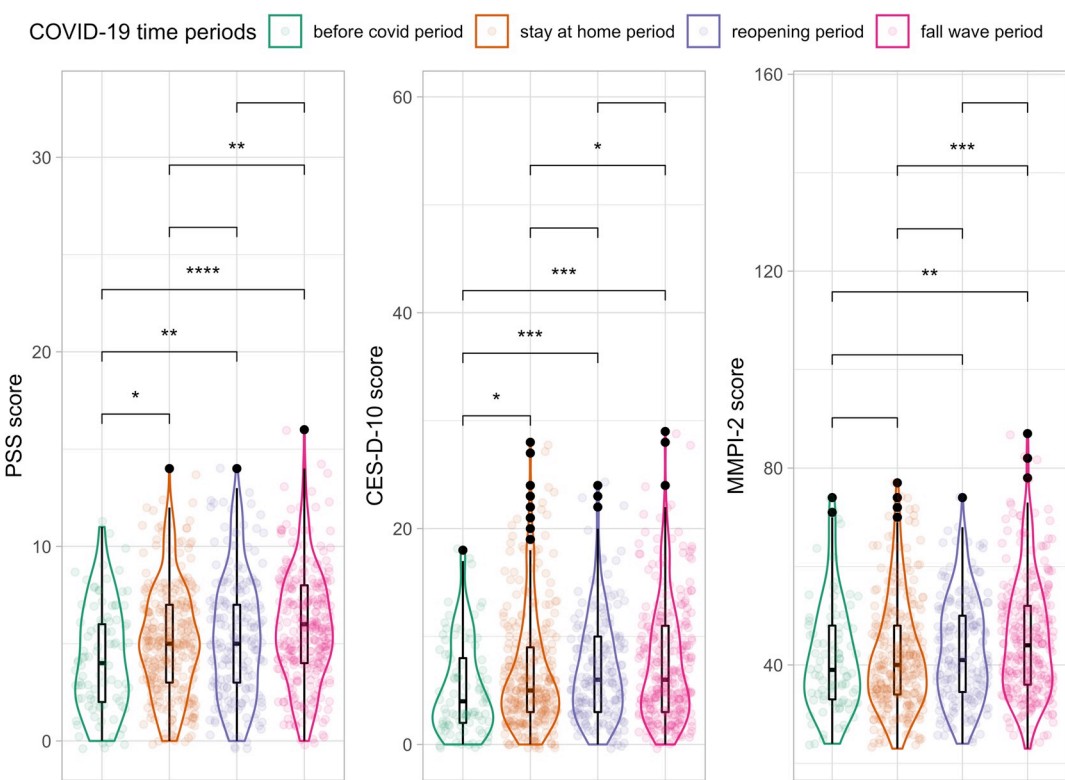

**Fig 2. Distributions for PSS-4, CES-D-10 depression, and MMPI-2 anxiety scores across COVID time periods N = 911.**
Significant differences between medians by time period for a given mental health metric was determined using a pairwise Wilcox Test. *: $p < = 0.05$, **: $p < = 0.01$, ***: $p < = 0.001$, ****: $p < = 0.0001$.

### Comparison of mental health measures by COVID-19 time period

We found some changes in average PSS-4, CES-D-10 depression, and MMPI-2 anxiety scores by time period (Fig 2). Depression and perceived stress scores in any of the phases of the pandemic in Denver were significantly higher compared to the "before covid" period. MMPI-2 anxiety scores were significantly higher in the "fall wave" compared to the "before covid" and "stay at home" periods. These differences could be true differences, or they could be due to different people responding to our survey during different time periods.

### Impacts of COVID-19 as predictors of stress, depression, and anxiety scores

During the COVID-19 pandemic, people faced significant stressors associated with the pandemic that may have affected their levels of stress and symptoms of depression and anxiety. In univariate analyses of COVID-19 impact variables on stress, anxiety, and depression scores (Table 2), COVID-19-related income losses and other financial impacts, difficulty obtaining resources like toilet paper or food, symptoms of COVID-19 –either in respondents themselves or others they spent time around–and substantial time spent online looking for information were each significantly associated with higher perceived stress, CES-D-10 depression, and MMPI-2 anxiety scores. Diagnosis with COVID-19 was not significantly associated with any of the mental health measures. Working in a job with higher levels of interpersonal contact and exposure was only significantly associated with higher anxiety scores and being around others who were diagnosed with or had symptoms of COVID-19 was borderline significantly associated with higher depression and anxiety scores (p<0.10). Self-report of spending more

**Table 2. Univariate regression of measures of psychological stress and distress and COVID-19 stressors.**

| COVID-19 variables | PSS Stress, N = 431 | | | CES-D-10 Depression, N = 427 | | | MMPI-2 Anxiety, N = 420 | | |
|---|---|---|---|---|---|---|---|---|---|
| | Beta | 95% CI | p-value | Beta | 95% CI | p-value | Beta | 95% CI | p-value |
| COVID-19 has impacted me negatively from a financial point of view | 0.35 | 0.23, 0.47 | <**0.001** | 0.33 | 0.12, 0.54 | **0.003** | 0.94 | 0.47, 1.4 | <**0.001** |
| I have lost job-related income due to COVID-19 | 0.23 | 0.12, 0.34 | <**0.001** | 0.23 | 0.04, 0.43 | **0.017** | 0.50 | 0.07, 0.93 | **0.022** |
| I have had a hard time getting needed resources (food, toilet paper) due to COVID-19 | 0.51 | 0.34, 0.68 | <**0.001** | 0.83 | 0.54, 1.1 | <**0.001** | 1.8 | 1.2, 2.5 | <**0.001** |
| I had to continue to work even though I was in close contact with people who might be infected (e.g., customers, patients, co-workers) | 0.08 | -0.04, 0.21 | 0.2 | 0.04 | -0.18, 0.25 | 0.7 | 0.70 | 0.23, 1.2 | **0.004** |
| I have been diagnosed with COVID-19 | | | | | | | | | |
| *Negative* | --- | --- | | --- | --- | | --- | --- | |
| *Positive* | -0.49 | -2.3, 1.3 | 0.6 | 0.79 | -2.3, 3.9 | 0.6 | 6.7 | -0.06, 13 | 0.053 |
| I have had coronavirus-like symptoms at some point in the last two months | 0.31 | 0.12, 0.49 | **0.001** | 0.73 | 0.41, 1.1 | <**0.001** | 2.1 | 1.4, 2.8 | <**0.001** |
| I have been in close proximity with someone who has had coronavirus-like symptoms in the last two months | 0.14 | 0.01, 0.27 | **0.031** | 0.39 | 0.16, 0.61 | <**0.001** | 0.83 | 0.33, 1.3 | **0.001** |
| I spend a huge percentage of my time trying to find updates online or on TV about COVID-19 | 0.30 | 0.13, 0.48 | <**0.001** | 0.41 | 0.10, 0.72 | **0.010** | 0.83 | 0.15, 1.5 | **0.018** |
| I have been in close proximity with someone who has been diagnosed with COVID-19 | 0.07 | -0.06, 0.20 | 0.3 | 0.22 | 0.00, 0.44 | 0.053 | 0.47 | -0.02, 1.0 | 0.062 |
| I have spent more time outside in parks, on trails, and near nature in the past month compared to the same time last year | -0.01 | -0.14, 0.12 | 0.9 | -0.12 | -0.35, 0.11 | 0.3 | 0.01 | -0.50, 0.52 | >0.9 |

time in green space during the pandemic compared to the same time in the previous year was not associated with any of the stress or distress scales we evaluated.

## Impact of green space on stress and mental health during the COVID-19 pandemic

In univariate analyses during the COVID-19 pandemic (i.e., excluding the "before covid" responses), all perceived and objective green space measures were associated with lower perceived stress, CES-D-10 depression, and MMPI-2 anxiety scores (S4 Table).

The associations of better mental health in greener areas were attenuated in analyses adjusted for sociodemographic variables (Table 3). Many of the perceived green space exposure metrics retained their significant negative (beneficial) associations with anxiety and depression scores, but most of them were no longer significantly associated with perceived stress.

Objective green space measures were no longer significantly associated with PSS-4 stress or MMPI-2 anxiety scores once adjusted for sociodemographic variables. NAIP NDVI was still associated with lower CES-D-10 depression scores after adjustment for sociodemographic variables at the 500m—but not 300m—radial buffer size. Overall, these findings were not sensitive to the choice of sociodemographic variables used in the adjustment. The only exception was that perceived stress became significantly associated with perceived greenspace visibility upon the addition of ethnicity to the sociodemographic adjustment variables (S5–S7 Tables).

## Impacts of green space on stress, anxiety, and depression adjusted for sociodemographic and COVID impact variables

Adding COVID-19 impact variables to the multivariate regressions further attenuated the relationships between green space and levels of stress, depression, and anxiety in our population

**Table 3. Associations between psychological stress and distress and perceived and objective green space, adjusted for sociodemographic variables.**

| | PSS Stress, N = 807[1] | | | CES-D-10 Depression, N = 801[2] | | | MMPI-2 Anxiety, N = 785[3] | | |
|---|---|---|---|---|---|---|---|---|---|
| Greenspace measure | Beta | 95% CI | p-value | Beta | 95% CI | p-value | Beta | 95% CI | p-value |
| "There is a lot of vegetation/greenery in my neighborhood" | | | | | | | | | |
| *Strongly Disagree* | — | — | | — | — | | — | — | |
| *Disagree* | -0.56 | -1.31, 0.20 | 0.149 | -1.54 | -2.91, -0.16 | **0.029** | -1.85 | -4.70, 1.00 | 0.204 |
| *Agree* | -0.42 | -1.10, 0.26 | 0.229 | -1.98 | -3.22, -0.74 | **0.002** | -2.72 | -5.29, -0.15 | **0.038** |
| *Strongly Agree* | -0.67 | -1.40, 0.06 | 0.073 | -2.60 | -3.94, -1.26 | **<0.001** | -5.11 | -7.89, -2.33 | **<0.001** |
| "I can see vegetation/greenery from my home" | | | | | | | | | |
| *Strongly Disagree* | — | — | | — | — | | — | — | |
| *Disagree* | -0.60 | -1.42, 0.22 | 0.153 | -0.40 | -1.90, 1.10 | 0.603 | -1.64 | -4.79, 1.52 | 0.310 |
| *Agree* | -0.61 | -1.33, 0.12 | 0.100 | -1.46 | -2.78, -0.14 | **0.031** | -2.73 | -5.52, 0.06 | 0.056 |
| *Strongly Agree* | -0.75 | -1.51, 0.01 | 0.054 | -1.86 | -3.25, -0.48 | **0.009** | -4.72 | -7.65, -1.79 | **0.002** |
| "The nearest vegetated park/green space is easy for me to access" | | | | | | | | | |
| *Strongly Disagree* | — | — | | — | — | | — | — | |
| *Disagree* | 0.05 | -1.25, 1.35 | 0.940 | -2.32 | -4.72, 0.08 | 0.058 | -4.46 | -9.37, 0.45 | 0.075 |
| *Agree* | -0.35 | -1.45, 0.74 | 0.527 | -2.77 | -4.77, -0.76 | **0.007** | -5.04 | -9.18, -0.91 | **0.017** |
| *Strongly Agree* | -0.69 | -1.78, 0.40 | 0.216 | -3.07 | -5.07, -1.06 | **0.003** | -7.25 | -11.38, -3.11 | **<0.001** |
| "I spend a lot of time in spaces with natural vegetation" | | | | | | | | | |
| *Strongly Disagree* | — | — | | — | — | | — | — | |
| *Disagree* | -0.48 | -1.30, 0.35 | 0.256 | -2.56 | -4.07, -1.05 | **<0.001** | -5.94 | -9.04, -2.83 | **<0.001** |
| *Agree* | -0.67 | -1.47, 0.13 | 0.099 | -3.34 | -4.79, -1.89 | **<0.001** | -7.97 | -10.98, -4.97 | **<0.001** |
| *Strongly Agree* | -0.91 | -1.75, -0.08 | **0.033** | -3.72 | -5.25, -2.20 | **<0.001** | -9.21 | -12.36, -6.06 | **<0.001** |
| "The green spaces near my home are very high quality" | | | | | | | | | |
| *Strongly Disagree* | — | — | | — | — | | — | — | |
| *Disagree* | -0.16 | -0.84, 0.52 | 0.640 | -0.81 | -2.05, 0.44 | 0.204 | -2.22 | -4.80, 0.36 | 0.092 |
| *Agree* | 0.02 | -0.62, 0.67 | 0.946 | -1.34 | -2.53, -0.15 | **0.027** | -3.76 | -6.23, -1.29 | **0.003** |
| *Strongly Agree* | -0.44 | -1.16, 0.28 | 0.236 | -2.31 | -3.63, -0.99 | **<0.001** | -6.07 | -8.82, -3.33 | **<0.001** |
| NAIP NDVI– 300 m buffer | 1.82 | -0.90, 4.54 | 0.191 | -4.37 | -9.38, 0.64 | 0.088 | -7.22 | -17.77, 3.34 | 0.181 |
| NAIP NDVI– 500 m buffer | 2.27 | -0.57, 5.12 | 0.118 | -5.41 | -10.65, -0.17 | **0.043** | -8.23 | -19.27, 2.82 | 0.145 |

[1]Adjusted for: sex, income, age, cohabitation status, educational attainment, employment status, insurance status

[2]Adjusted for: income, age, cohabitation status, educational attainment, employment status, insurance status

[3]Adjusted for: income, ethnicity, age, cohabitation status, educational attainment, employment status, insurance status

(Table 4). None of the green space measures, whether perceived or objective, was significantly associated with PSS-4 stress scores after adjustment for sociodemographic and COVID-19 impact variables. Self-report of spending a lot of time in green space (usage) was still significantly associated with lower depression and anxiety scores for individuals reporting 'Strongly agree' or 'Agree' to those questions compared to those who responded 'Strongly Disagree'. Self-report of a lot of green space in one's neighborhood (abundance) and that the green space in one's neighborhood was of high quality (quality) were each only significantly associated with lower MMPI-2 anxiety scores for those who responded 'Strongly Agree' compared to those who responded 'Strongly Disagree' with adjustment for COVID-19 impact variables and sociodemographic variables. Seeing greenery from one's home (visibility) and being able to access green space easily (access) were no longer significantly associated with lower scores on the CES-D-10 depression or MMPI-2 anxiety scales once COVID-19 impacts scores were additionally adjusted for. The objective green space metric remained significantly associated with lower CES-D-10 depression scores, but this time at both buffer scales.

**Table 4. Multivariate regression of measures of psychological stress and distress and perceived and objective green space, controlling for sociodemographic and COVID-19 variables.**

| Greenspace measure | PSS Stress, N = 807[1] | | | CES-D-10 Depression, N = 801[2] | | | MMPI-2 Anxiety, N = 785[3] | | |
|---|---|---|---|---|---|---|---|---|---|
| | Beta | 95% CI | p-value | Beta | 95% CI | p-value | Beta | 95% CI | p-value |
| "There is a lot of vegetation/greenery in my neighborhood" | | | | | | | | | |
| *Strongly Disagree* | — | — | | — | — | | — | — | |
| *Disagree* | 0.01 | -0.99, 1.01 | 0.990 | -0.21 | -1.98, 1.56 | 0.818 | 0.24 | -3.52, 4.00 | 0.900 |
| *Agree* | -0.21 | -1.11, 0.68 | 0.639 | -1.02 | -2.61, 0.57 | 0.209 | -1.38 | -4.74, 1.99 | 0.423 |
| *Strongly Agree* | -0.41 | -1.38, 0.57 | 0.414 | -1.48 | -3.21, 0.24 | 0.093 | -4.03 | -7.69, -0.37 | **0.032** |
| "I can see vegetation/greenery from my home" | | | | | | | | | |
| *Strongly Disagree* | — | — | | — | — | | — | — | |
| *Disagree* | -0.18 | -1.28, 0.91 | 0.743 | 1.24 | -0.68, 3.17 | 0.206 | 1.02 | -3.13, 5.16 | 0.630 |
| *Agree* | -0.04 | -1.00, 0.92 | 0.932 | 0.37 | -1.31, 2.05 | 0.665 | 2.21 | -1.44, 5.86 | 0.237 |
| *Strongly Agree* | -0.33 | -1.33, 0.67 | 0.521 | -0.48 | -2.23, 1.28 | 0.595 | -1.55 | -5.37, 2.26 | 0.426 |
| "The nearest vegetated park/green space is easy for me to access" | | | | | | | | | |
| *Strongly Disagree* | — | — | | — | — | | — | — | |
| *Disagree* | 0.46 | -1.52, 2.43 | 0.650 | 0.18 | -3.30, 3.66 | 0.918 | -1.26 | -8.67, 6.15 | 0.739 |
| *Agree* | 0.30 | -1.23, 1.82 | 0.704 | 0.92 | -1.76, 3.61 | 0.501 | -0.54 | -6.25, 5.17 | 0.854 |
| *Strongly Agree* | -0.01 | -1.55, 1.53 | 0.991 | 0.54 | -2.17, 3.24 | 0.698 | -2.48 | -8.23, 3.27 | 0.398 |
| "I spend a lot of time in spaces with natural vegetation" | | | | | | | | | |
| *Strongly Disagree* | — | — | | — | — | | — | — | |
| *Disagree* | -0.07 | -1.18, 1.04 | 0.902 | -0.62 | -2.59, 1.35 | 0.538 | -1.65 | -5.77, 2.47 | 0.434 |
| *Agree* | -0.78 | -1.84, 0.28 | 0.149 | -2.23 | -4.11, -0.35 | **0.020** | -5.76 | -9.68, -1.84 | **0.004** |
| *Strongly Agree* | -0.92 | -2.03, 0.19 | 0.106 | -2.35 | -4.31, -0.39 | **0.020** | -6.51 | -10.62, -2.40 | **0.002** |
| "The green spaces near my home are very high quality" | | | | | | | | | |
| *Strongly Disagree* | — | — | | — | — | | — | — | |
| *Disagree* | -0.40 | -1.33, 0.53 | 0.396 | 0.02 | -1.63, 1.67 | 0.985 | -2.16 | -5.67, 1.35 | 0.228 |
| *Agree* | 0.14 | -0.73, 1.02 | 0.748 | -0.08 | -1.63, 1.46 | 0.918 | -2.19 | -5.48, 1.10 | 0.193 |
| *Strongly Agree* | -0.42 | -1.41, 0.57 | 0.405 | -1.68 | -3.43, 0.07 | 0.060 | -6.17 | -9.88, -2.46 | **0.001** |
| NAIP NDVI– 300 m buffer | -0.10 | -3.93, 3.74 | 0.961 | -7.09 | -13.82, -0.37 | **0.039** | -10.94 | -25.55, 3.66 | 0.143 |
| NAIP NDVI– 500 m buffer | 0.80 | -3.22, 4.83 | 0.696 | -7.78 | -14.85, -0.71 | **0.032** | -12.22 | -27.54, 3.09 | 0.119 |

[1]Adjusted for: sex, income, age, cohabitation status, educational attainment, employment status, insurance status, and COVID-19 related: financial stress, lost income, resource scarcity, symptoms, close proximity to symptoms, and online engagement

[2]Adjusted for: income, age, cohabitation status, educational attainment, employment status, insurance status, and COVID-19 related: financial stress, lost income, resource scarcity, symptoms, close proximity to symptoms, and online engagement

[3]Adjusted for: income, ethnicity, age, cohabitation status, educational attainment, employment status, insurance status, and COVID-19 related: financial stress, lost income, resource scarcity, potential workplace exposure, symptoms, close proximity to symptoms, and online engagement

## Discussion

Our study documents that both perceived and objective green space measures were associated with lower depression and anxiety scores–but not perceived stress scores–during the COVID-19 pandemic among survey respondents in Denver, CO, even after adjusting for potential socio-economic confounding factors within our cross-sectional survey. This corroborates many other studies that document the benefits of green space on mental health during the COVID-19 pandemic [14, 15, 19, 20, 23–25, 61]. This is particularly important given the mental health impacts of the pandemic have been well documented [1–8, 11, 62], including in our study (Fig 2).

Our study, however, extends the knowledge of how green space may have benefited mental health during the COVID-19 pandemic in a few ways. Although other studies investigating the

role of green space on mental health during the pandemic adjusted for COVID-19 related changes to household income [23, 25, 61] and the number of COVID-19 cases [23], we did not find any studies that additionally adjusted for the range of impacts of the pandemic that likely affected mental health, such as financial and health concerns, that we did. Importantly, once we additionally adjusted for stressors known to occur during the COVID-19 pandemic, many of the significant associations between green space and lower depression and anxiety scores were no longer significant, but the perceived metric of reporting spending more time in green space ("spaces with natural vegetation") remained significantly associated with lower depression and anxiety scores. There was also an association between strongly agreeing about the quality of green spaces near one's home and lower anxiety levels during the COVID-19 pandemic. Only one objective measure of green space (NAIP NDVI) was associated with lower depression scores, and no objective measures were associated with lower anxiety scores after adjustment for stressors of the pandemic.

Another benefit of our study was that we investigated how different aspects of green space exposure–measured by our five questions of perceived green space and one measure of objective green space–affected mental health during the pandemic. The five questions we used about perceived green space investigated various dimensions of green space exposure such as abundance, visibility, access, usage, and quality, whereas most of the previous studies investigating the role of green space on mental health during COVID-19 have focused on just one or two of these. For example, most focus on usage and visibility [23, 61], just usage [14, 19, 20], or visibility and access [25], visibility and abundance [24], or access, abundance, usage [17].

Although one should not infer causality from a cross-sectional study, our findings appear to show that spending time in green space (usage) may have conferred benefits for depression and anxiety and that higher perceived quality of nearby green space was beneficial for anxiety, adjusted for sociodemographic and COVID-19 impact variables. We did observe lower mental health scores associated with other perceived exposure to green space measures such as abundance, visibility, or access but only when not adjusted for impacts of the COVID-19 pandemic which could affect mental health. Other studies have found benefits to mental health during the pandemic for visibility [23–25, 61], abundance [17, 19], and access [25], and usage [14, 20, 23, 61]. It is worth mentioning, however, that unlike the other studies, our models were adjusted for a whole host of COVID-19 impact variables, which attenuated many significant relationships that we observed in models adjusted for only sociodemographic variables. In these models adjusted only for sociodemographic variables, we found significant associations between green space abundance, visibility, access, usage, and quality for both depression and anxiety, and between green space usage and stress.

The differences in findings across studies could be due to numerous factors such as differences in the public health orders about what was allowed (i.e., in some locations, one could not use public green spaces, which could make visibility more important), differences in the context of green spaces available (i.e., possibly in cities with less overall green space compared to Denver, visibility or abundance is more important), or the way that the questions about green space exposure were worded. We note that many previous studies asked questions that directly linked green space exposure and mental health, which could lead to confirmation bias. Our survey intentionally placed the mental health questions before the green space exposure questions in our survey such that the respondents would be less likely to respond to the mental health questions while thinking about their green space exposure. Additionally, our COVID-19 impact questions were the last questions in the survey, such that the mental health and green space questions were not primed by thinking about the pandemic, even though it was likely present in everyone's minds in 2020. Given that we found no other studies that investigated perceived quality of green space on mental health and that we found this was associated

with lower anxiety scores, we recommend more studies investigate perceived quality as one way that green space may benefit mental health.

Despite the fact that our fully adjusted model and others' studies [17, 19, 24] have not found significant associations between perceived abundance of green space and mental health during the pandemic, we did find a relationship between greenspace abundance measured by NDVI and improved mental health outcomes, which several other studies have observed as well [20]. We found that objective green space, as measured by NDVI within 300m and 500m of a participant's address, was associated with lower depression scores during the pandemic adjusted for sociodemographic and COVID-19 impact variables. One might consider a measure of the quantity of vegetation nearby (which NDVI is a proxy for) as another indicator of abundance., Although there was some correlation between these measures (r = 0.59), it was not an overwhelmingly high correlation, which may explain the divergence in our findings of these exposures with mental health metrics. Further research is needed to understand why people's perceptions of abundance do not agree with objective measures of nearby vegetation. This could be very important given the preponderance of studies that rely upon NDVI as a metric of green space exposure in health studies [13].

We did not observe any significant associations between stress (as measured by the PSS scale) and green space in adjusted models despite the benefits for anxiety and depression for some green space exposure metrics during the pandemic in our study population. Robinson et al. (2021) also found no significant benefit for NDVI within a variety of radial buffers on PSS scores during the pandemic, but they did find that self-reported duration and frequency of time in nature were significantly associated with lower PSS scores. The difference in findings from their study to ours could be due to the different measures of perceived green space exposure, the differences in green space patterns and pandemic impacts in the contexts of these studies (their respondents mostly lived in England and ours in Denver, CO, USA), or other aspects of study design.

Among our survey respondents, 32.9% reported spending more time in parks, on trails, and in nature during the pandemic than before the pandemic, which is similar to 27.6% in a study in Australia [17], but much less than the 88% increase found in a study mostly in England [20]. On the other hand, some studies have documented decreased exposure to nature during the pandemic [14, 16]. It is not clear if the differences in findings across studies is due to different public health measures in different places or how the questions were worded, which could elicit different responses. At least one study documented that the different associations between specific green space exposures and mental health during the pandemic for Spain and Portugal could be due to how impacted each of those countries were by COVID-19 and how long lockdown orders remained in effect and whether public parks were closed or not [20]. In a study in South Korea, if people noted that they decreased their green space visits, they were then asked why, and many people responded that they were fearful about catching SARS-CoV-2 from others, that there was increased crowding in green spaces, or that the government had either closed green spaces or they were following government orders to not leave their homes [14]. We note that our question was worded to ask whether the participants used green space more during the pandemic than before, but that does not allow us to infer that people who responded on the low end of our scale decreased their time in nature during the pandemic or whether it remained the same. We suggest that future research try to disentangle to what extent the differences in reported behaviors related to green spaces are due to the wording of questions by researchers or to the policies enacted during the pandemic.

Our study also documents which stressors of the pandemic itself were most associated with measures of stress, anxiety, and depression. We observed the strongest associations with higher stress, depression, and anxiety scores with not being able to access needed resources (such as

toilet paper) during the pandemic. These findings underscore the need to better understand which resources produced the most stress to better understand how to ensure that populations have access to what they need in future public health and other crises. Interestingly, having been diagnosed with COVID-19 was not associated with stress or mental health scores, but having experienced COVID-19 symptoms was. This could imply that the potential of having COVID-19 was more stressful than actually having been diagnosed, or it could be due to the stress and anxiety of having symptoms during times when testing was hard to find. Regardless of the metric (i.e., PSS-4, CES-D-10 depression, or MMPI-2 anxiety), people who lost job-related income or were otherwise financially impacted by the pandemic had higher mental health and stress scores. Additionally, respondents who reported spending a lot of time online looking for updates about the pandemic reported higher stress and mental health scores. This corroborates findings from past studies that have found associations between COVID-19-related news and media coverage and stress, anxiety, and depression [5, 11, 63]; among these studies, several indicate that the negative association between COVID-19-related media and mental health indicators may be due to misinformation or conflicting information [5, 11]. Another interesting, but possibly not surprising, finding was that having to work in close contact with others who may have symptoms of COVID-19 was associated with higher anxiety, but not depression or stress. This understanding is important to document in its own right but is also helpful for our ongoing work using this survey to understand the mental health impacts of green space exposure given that many of our survey responses occurred during the pandemic.

This study has some limitations that are worth noting. Although our study has responses from various times in the pandemic, we only have one response per person, thus our study is cross-sectional which limits our ability to consider the results causal. Future work should be done longitudinally with repeated measures from participants to understand if changes in objective or perceived exposures to green space change mental health metrics. Similarly, because the study focused on respondents from one city, the findings here should not be considered to apply in all locations, especially given how the pandemic and associated public health measures differed throughout the world. We note, however, that many studies have documented benefits of green space on mental health during the stressful time of the COVID-19 pandemic in many places throughout the world, such that our findings may be more universal than just this population in Denver, CO, USA. Our study population differs from the city of Denver in key demographic variables which we adjusted for in our analyses but does limit the generalizability of the findings. There is also the potential that our findings suffer from self-selection bias, whereby people with better mental health are more likely to choose to live in greener areas. Because our study is cross-sectional, we could not look at how people choose to move related to these characteristics, but we did adjust for factors that are known to be associated with differential exposure to green space such as income, educational attainment, and race/ethnicity. Although some of our findings focused on a self-reported exposure (our perceived exposure measures) and a self-reported outcome, which could cause bias, none of the questions in our survey linked green space and mental health into the same question like many of the other studies cited in this document [15–17, 20], which we think could lead to confirmation bias more readily. Additionally, we ordered our survey questions so that the mental health questions were answered before the green space exposure question to attempt to limit the respondent from guessing that we hypothesized a link between green space exposure and mental health.

Our research provides evidence to support municipal policies that do more than just add more green space (abundance), but that also work to improve the perceived quality and usage of green space by residents. Working with community groups to determine what denotes

better quality of green space and what makes people more likely to use green space is an important area of future research. Given the growing evidence of mental health benefits of green space [13] and the growing evidence of benefits of green space for mental health during the pandemic highlighted by this and other studies [14, 20, 23–25], it is essential that policies are enacted to ensure that green spaces are available to residents during the rest of this pandemic and during future pandemics or other societal challenges. Additionally, our research adds to a growing body of evidence, which includes evidence of plausible biological mechanisms and from experimental studies [64], documenting mental health benefits of green space generally.

## Conclusion

This is the first study, to our knowledge, to analyze the role of both perceived and objective green space on health during the COVID-19 pandemic that also took into account many of the stressors associated with the COVID-19 pandemic. Additionally, our study provided more nuance to our understanding of green space exposure implying that spending more time in green space in addition to having it nearby was particularly associated with lower depression scores, whereas perceiving higher quality green space was associated with lower anxiety scores, when adjusted for the specific stressors that the global COVID-19 pandemic imparted as well as sociodemographic variables. Municipal governments should work to increase amounts of and use of green spaces as a public health measure to improve mental health.

## Supporting information

**S1 File. Survey questions.**
(PDF)

**S1 Table. Associations between measures of psychological stress and distress and sociodemographic characteristics in univariate regressions.**
(DOCX)

**S2 Table. Demographic comparison of the study population to the denver population.**
(DOCX)

**S3 Table. Descriptive statistics for COVID-19 impact questions.**
(DOCX)

**S4 Table. Associations between measures of psychological stress and distress and perceived and objective green space in univariate regressions.**
(DOCX)

**S5 Table. PSS sensitivity analysis results.**
(DOCX)

**S6 Table. CES-D sensitivity analysis results.**
(DOCX)

**S7 Table. MMPI sensitivity analysis results.**
(DOCX)

## Acknowledgments

We acknowledge the work of Liyu Berhanu and Julianna Rohn for their extensive administrative work to support recruitment of survey participants.

## Author Contributions

**Conceptualization:** Colleen E. Reid.

**Data curation:** Kate Carlson.

**Formal analysis:** Emma S. Rieves.

**Funding acquisition:** Colleen E. Reid.

**Methodology:** Colleen E. Reid.

**Project administration:** Colleen E. Reid.

**Visualization:** Emma S. Rieves.

**Writing – original draft:** Colleen E. Reid, Emma S. Rieves.

**Writing – review & editing:** Colleen E. Reid, Emma S. Rieves.

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
