## [Decision Letter · Decision Letter 0]

14 Sep 2021

PONE-D-21-23016Usage, abundance, and quality of green space benefit mental health during the COVID-19 pandemicPLOS ONE

Dear Dr. Reid,

Thank you for submitting your manuscript to PLOS ONE. After careful consideration, we feel that it has merit but does not fully meet PLOS ONE’s publication criteria as it currently stands. Therefore, we invite you to submit a revised version of the manuscript that addresses the points raised during the review process. Both the referees raise questions about the problem of sample selection. It is possible that respondents who live near (or have access to) green spaces do better on mental health and hence are able to afford green spaces (or better living conditions) in the first place. While you may not be able to solve for causality, I would suggest that a) you spend more time describing the location of your sample (perhaps some maps of Denver and how the green spaces and mental health results appear on the maps, and b) nuances your piece so that you are not making any causal claims, and are only reporting a correlation. Please submit your revised manuscript by Oct 29 2021 11:59PM. If you will need more time than this to complete your revisions, please reply to this message or contact the journal office at plosone@plos.org. Please include the following items when submitting your revised manuscript:A rebuttal letter that responds to each point raised by the academic editor and reviewer(s). You should upload this letter as a separate file labeled 'Response to Reviewers'.A marked-up copy of your manuscript that highlights changes made to the original version. You should upload this as a separate file labeled 'Revised Manuscript with Track Changes'.An unmarked version of your revised paper without tracked changes. You should upload this as a separate file labeled 'Manuscript'.

We look forward to receiving your revised manuscript.

Kind regards,

Renuka Sane

Academic Editor

PLOS ONE

Journal Requirements:

"The CUPC is supported by funds from the Eunice Kennedy Shriver National Institute of Child Health & Human Development of the National Institutes of Health (NIH) under Award Number P2CHD066613. The content is solely the responsibility of the authors and does not necessarily represent the official views of the NIH"

"Research reported in this publication was supported by the Harvard JPB Environmental Health Fellowship (CER) and the Developmental Core of the University of Colorado Population Center (CUPC) (CER). The CUPC is supported by funds from the Eunice Kennedy Shriver National Institute of Child Health & Human Development of the National Institutes of Health (NIH) under Award Number P2CHD066613. The content is solely the responsibility of the authors and does not necessarily represent the official views of the NIH. NIH and CUPC did not play a role in the study design, data collection and analysis, decision to publish, or preparation of the manuscript. Researchers at Harvard provided feedback on the study design, but the JPB funders did not. Neither JPB nor Harvard played a role in the data collection and analysis, decision to publish, or preparation of the manuscript."

Reviewers' comments:

Reviewer's Responses to Questions

**Comments to the Author**

1. Is the manuscript technically sound, and do the data support the conclusions?

Reviewer #1: Partly

Reviewer #2: Yes

2. Has the statistical analysis been performed appropriately and rigorously? 

Reviewer #1: Yes

Reviewer #2: Yes

3. Have the authors made all data underlying the findings in their manuscript fully available?

Reviewer #1: No

Reviewer #2: Yes

4. Is the manuscript presented in an intelligible fashion and written in standard English?

Reviewer #1: Yes

Reviewer #2: Yes

5. Review Comments to the Author

Reviewer #1: This paper uses survey results to further our understanding of the potential for green space to provide urban residents with mental health benefits. Especially in the context of the COVID-19 pandemic this is an important topic that the authors have much to contribute to. The inclusion of multiple measures of green space, multiple mental health measures, and pandemic related stressors presents and improvement on previous research that fails to account for this wide range of influences and impacts.

However, there remain several major issues that need to be addressed before the manuscript is suitable for publication, specifically:

• The survey results are compared between the different time periods, but with different respondents in each period, it might simply be a result of who is responding rather than any wider differences among Denver residents

• There is no comparison of the respondents (which seem to skew whiter and wealthier) with Denver’s population. Which do the results speak to?

o Similarly, I didn’t see a response rate anywhere in the MS (apologies if I missed it)

• The policy implications need to be more developed-what can policymakers and managers take away from this research?

Other less pressing issues include:

• The single sentence paragraph at the end of the discussion section (Lines 591-593) should be integrated with previous paragraph or cut

• It would be useful for the reader unfamiliar with Denver’s park system to get a description of the city’s greenspace and who has access to it

• A justification of the distance buffers used to measure “objective” green space exposure would be useful-even just citations of previous studies that use similar distances

• The survey delivery method is introduced (Lines 221-222) before the survey is finished being described, I would move it down after the perceived green space paragraph/section

Reviewer #2: Fundamentally it is a good paper. It is an important question and the authors have nicely marshalled a unique set of data to offer insights.

I have one main concern, and that is about self-selection. Humans with certain attributes would tend to self-select themselves to live in places with greater access to vegetation. These attributes could easily be correlated with some of the phenomena of interest. Thus, this estimation strategy runs the risk of endogeneity bias. As an example, there's a well known correlation between physics researchers and the mountains.

The research strategy adopted by the authors will inevitably face these constraints. The writing of the paper should be more careful to stop short of making causal claims, and raise the concern that these correlations could be in fray.

6. PLOS authors have the option to publish the peer review history of their article (what does this mean?). If published, this will include your full peer review and any attached files.

Reviewer #1: No

Reviewer #2: No

---

## [Author Response · Author response to Decision Letter 0]

3 Dec 2021

Response to Reviewers

We have revised our manuscript to respond to the edits of the reviewers and editor and we thank them for their thoughtful comments that we believe have improved out paper. Our responses to each comment are below in italics below each comment. Additionally, we document where in the manuscript the reviewer can find any associated changes in the text. The line numbers refer to the line numbers of the word document with tracked changes. 

Additionally, we have placed all of the data in a repository by our university and will soon be receiving the doi for that data that we can provide to Plos One to include with our manuscript. 

Reviewer #1: This paper uses survey results to further our understanding of the potential for green space to provide urban residents with mental health benefits. Especially in the context of the COVID-19 pandemic this is an important topic that the authors have much to contribute to. The inclusion of multiple measures of green space, multiple mental health measures, and pandemic related stressors presents and improvement on previous research that fails to account for this wide range of influences and impacts.

Thank you for this comment. 

However, there remain several major issues that need to be addressed before the manuscript is suitable for publication, specifically:

• The survey results are compared between the different time periods, but with different respondents in each period, it might simply be a result of who is responding rather than any wider differences among Denver residents

We agree that this is possible and have added in the following “These differences could be true differences or they could be due to different people responding to our survey during different time periods.” on lines 405-406 in the edited manuscript. 

• There is no comparison of the respondents (which seem to skew whiter and wealthier) with Denver’s population. Which do the results speak to?

Thank you for pointing this out. We have added in a supplemental table (S2 Table) that shows our population percentages by demographic characteristics compared to Denver as a whole and the neighborhoods that were targeted in our initial screening. We additionally state in the results section on lines 352-356: 

“Respondents to our survey were more likely to be female, older, White, and not Hispanic/Latino than the population of Denver or the neighborhoods in Denver that we targeted. They were also more likely to have a Bachelor’s degree, and a higher income than the average in Denver or the neighborhoods we targeted, but they were about equally likely to have health insurance as the population of Denver (S2 Table). ”

We also added the following related sentence to our discussion section on lines 641-643: “Our study population differs from the city of Denver in key demographic variables which we adjusted for in our analyses but does limit the generalizability of the findings.” 

o Similarly, I didn’t see a response rate anywhere in the MS (apologies if I missed it)

Thank you for pointing out this omission. We have added in our response rate and completion rate into the section on the study population on lines 340-341.

• The policy implications need to be more developed-what can policymakers and managers take away from this research?

Thank you. We have put in a few more sentences at the end of the discussion about policy implications of this and other work on the mental health benefits of green spaces particularly during the pandemic.

Other less pressing issues include:

• The single sentence paragraph at the end of the discussion section (Lines 591-593) should be integrated with previous paragraph or cut

Thank you – we have added more to this paragraph on policy implications which deals with this issue as well as the previous comment. This section is now on lines 655-665 and reads:

“Our research provides evidence to support municipal policies that do more than just add more green space (abundance), but that also work to improve the perceived quality and usage of green space by residents. Working with community groups to determine what denotes better quality of green space and what makes people more likely to use green space is an important area of future research. Given the growing evidence of mental health benefits of green space [13] and the growing evidence of benefits of green space for mental health during the pandemic highlighted by this and other studies [14,20,23–25], it is essential that policies are enacted to ensure that green spaces are available to residents during the rest of this pandemic and during future pandemics or other societal challenges. Additionally, our research adds to a growing body of evidence, which includes evidence of plausible biological mechanisms and from experimental studies [64], documenting mental health benefits of green space generally.”

• It would be useful for the reader unfamiliar with Denver’s park system to get a description of the city’s greenspace and who has access to it

We have added a section on the park system and what is known about it in the “Study Population” section found on lines 143-145. This sentence says: “Denver is an interesting city from which to study green space exposure and health because of previous studies documenting unequal access to parks by race/ethnicity in Denver [42] and lower levels of vegetation, as measured by satellites, in lower income areas of Denver [43].”

• A justification of the distance buffers used to measure “objective” green space exposure would be useful-even just citations of previous studies that use similar distances

We have added to lines 292-296 this statement, “There is no agreement on what radial buffer size is most appropriate in green space and health studies. Different studies have found that both larger and smaller buffers are better, but despite this disagreement, the clearest evidence is that the buffer size matters [55,56]. We chose these buffer sizes because they are commonly used in green space and mental health research [57].”

• The survey delivery method is introduced (Lines 221-222) before the survey is finished being described, I would move it down after the perceived green space paragraph/section

Thank you, we have done this. 

Reviewer #2: Fundamentally it is a good paper. It is an important question and the authors have nicely marshalled a unique set of data to offer insights.

I have one main concern, and that is about self-selection. Humans with certain attributes would tend to self-select themselves to live in places with greater access to vegetation. These attributes could easily be correlated with some of the phenomena of interest. Thus, this estimation strategy runs the risk of endogeneity bias. As an example, there's a well known correlation between physics researchers and the mountains.

Thank you for this comment. We agree that this is a potential concern that individuals with better mental health may choose to live in greener areas, which could indicate a self-selection bias that would lead to reverse causality whereby mental health is causing green space exposure rather than green space exposure causing better mental health. This self-selection bias, to our knowledge, has not been assessed for green space exposure and mental health outcomes, although at least one study disproved it for green space exposure and physical activity or obesity measures (4). 

One of the ways that we adjusted for the potential that people with certain characteristics may be more likely to live in places with more vegetation was to adjust for factors that we collected that could be such factors such as income and educational attainment. We have also now addressed this concern in our discussion section on lines 641-645 as follows:

“There is also the potential that our findings suffer from self-selection bias, whereby people with better mental health are more likely to choose to live in greener areas. Because our study is cross-sectional, we could not look at how people choose to move related to these characteristics, but we did adjust for factors that are known to be associated with differential exposure to green space such as income, educational attainment, and race/ethnicity.”

The research strategy adopted by the authors will inevitably face these constraints. The writing of the paper should be more careful to stop short of making causal claims, and raise the concern that these correlations could be in fray.

Thank you for this. We have reread and edited our paper to ensure that we are not making causal statements. We already had this statement in our discussion: “Although our study has responses from various times in the pandemic, we only have one response per person, thus our study is cross-sectional which limits our ability to consider the results causal” (on lines 629-631 now), but have now made editorial changes – most notably to our title – to ensure that we are not overstating our findings. 

We have reviewed our formatting and believe that we have made the necessary changes to align our manuscript with the style templates provided.

"The CUPC is supported by funds from the Eunice Kennedy Shriver National Institute of Child Health & Human Development of the National Institutes of Health (NIH) under Award Number P2CHD066613. The content is solely the responsibility of the authors and does not necessarily represent the official views of the NIH"

"Research reported in this publication was supported by the Harvard JPB Environmental Health Fellowship (CER) and the Developmental Core of the University of Colorado Population Center (CUPC) (CER). The CUPC is supported by funds from the Eunice Kennedy Shriver National Institute of Child Health & Human Development of the National Institutes of Health (NIH) under Award Number P2CHD066613. The content is solely the responsibility of the authors and does not necessarily represent the official views of the NIH. NIH and CUPC did not play a role in the study design, data collection and analysis, decision to publish, or preparation of the manuscript. Researchers at Harvard provided feedback on the study design, but the JPB funders did not. Neither JPB nor Harvard played a role in the data collection and analysis, decision to publish, or preparation of the manuscript."

We have removed the funding statement from the text and have updated the funding statement in the resubmission. 

References

1. Reid CE, Kubzansky LD, Li J, Shmool JL, Clougherty JE. It’s not easy assessing greenness: A comparison of NDVI datasets and neighborhood types and their associations with self-rated health in New York City. Health Place. 2018 Sep 21;54:92–101. 

2. James P, Berrigan D, Hart JE, Hipp JA, Hoehner CM, Kerr J, et al. Effects of buffer size and shape on associations between the built environment and energy balance. Health & place. 2014 May;27:162–70. 

3. Davis Z, Guhn M, Jarvis I, Jerrett M, Nesbitt L, Oberlander T, et al. The association between natural environments and childhood mental health and development: A systematic review and assessment of different exposure measurements. Int J Hyg Environ Health. 2021 May 11;235:113767. 

4. James P, Hart JE, Arcaya MC, Feskanich D, Laden F, Subramanian SV. Neighborhood Self-Selection: The Role of Pre-Move Health Factors on the Built and Socioeconomic Environment. International Journal of Environmental Research and Public Health. 2015 Oct;12(10):12489–504.

---

## [Decision Letter · Decision Letter 1]

27 Jan 2022

Perceptions of green space usage, abundance, and quality of green space were associated with better mental health during the COVID-19 pandemic among residents of Denver

PONE-D-21-23016R1

Dear Dr. Reid,

We’re pleased to inform you that your manuscript has been judged scientifically suitable for publication and will be formally accepted for publication once it meets all outstanding technical requirements.

Kind regards,

Renuka Sane

Academic Editor

PLOS ONE

Additional Editor Comments (optional):

Reviewers' comments:

Reviewer's Responses to Questions

**Comments to the Author**

1. If the authors have adequately addressed your comments raised in a previous round of review and you feel that this manuscript is now acceptable for publication, you may indicate that here to bypass the “Comments to the Author” section, enter your conflict of interest statement in the “Confidential to Editor” section, and submit your "Accept" recommendation.

Reviewer #1: All comments have been addressed

2. Is the manuscript technically sound, and do the data support the conclusions?

Reviewer #1: Yes

3. Has the statistical analysis been performed appropriately and rigorously? 

Reviewer #1: Yes

4. Have the authors made all data underlying the findings in their manuscript fully available?

Reviewer #1: Yes

5. Is the manuscript presented in an intelligible fashion and written in standard English?

Reviewer #1: Yes

6. Review Comments to the Author

Reviewer #1: Many thanks to the authors for their time spent responding to my comments and concerns regarding this manuscript. I am satisfied with their responses and happy to recommend publication.

7. PLOS authors have the option to publish the peer review history of their article (what does this mean?). If published, this will include your full peer review and any attached files.

Reviewer #1: No

---

## [Editor Report · Acceptance letter]

8 Feb 2022

PONE-D-21-23016R1 

Perceptions of green space usage, abundance, and quality of green space were associated with better mental health during the COVID-19 pandemic among residents of Denver 

Dear Dr. Reid:

I'm pleased to inform you that your manuscript has been deemed suitable for publication in PLOS ONE. Congratulations! Your manuscript is now with our production department. 

Kind regards, 

on behalf of

Dr. Renuka Sane 

Academic Editor

PLOS ONE